# Retrospective Trial on Cetuximab Plus Radiotherapy in Elderly Patients with Head and Neck Squamous Cell Cancer

**DOI:** 10.3390/cancers17213550

**Published:** 2025-11-02

**Authors:** Morena Fasano, Francesco Perri, Mario Pirozzi, Chiara Lucrezia Deantoni, Davide Valsecchi, Alessio Cirillo, Raffaele Addeo, Pasquale Vitale, Francesca De Felice, Paolo Tralongo, Stefano Farese, Beatrice Ruffilli, Fabrizio Romano, Mathilda Guizzardi, Leone Giordano, Monica Pontone, Maria Luisa Marciano, Fabiana Raffaella Rampetta, Francesco Longo, Fortunato Ciardiello, Aurora Mirabile

**Affiliations:** 1Division of Medical Oncology, Department of Precision Medicine, University of Campania Luigi Vanvitelli, Via Sergio Pansini 5, 80131 Naples, Italy; morena.fasano@unicampania.it (M.F.); stefano.farese@studenti.unicampania.it (S.F.); fortunato.ciardiello@unicampania.it (F.C.); 2Head and Neck Oncology Unit, Istituto Nazionale Tumori di Napoli IRCCS “G. Pascale”, Via Semmola 53, 80131 Naples, Italy; m.pontone@istitutotumori.na.it (M.P.); ml.marciano@istitutotumori.na.it (M.L.M.); fabianaraffaella.rampetta@istitutotumori.na.it (F.R.R.); 3Department of Translational Medicine (DIMET), University of Eastern Piedmont (UPO), 28100 Novara, Italy20023658@studenti.uniupo.it (B.R.); 4SCDU Oncologia, AOU Maggiore della Carità di Novara, 28100 Novara, Italy; 5Radiation Oncology Department, IRCCS San Raffaele Scientific Institute, 20132 Milan, Italy; deantoni.chiaralucrezia@hsr.it; 6Emergency Department, IRCCS San Raffaele Scientific Institute, 20132 Milan, Italy; valsecchi.davide@hsr.it; 7Department of Radiological, Oncological and Pathological Sciences, Sapienza University, 00161 Rome, Italy; alessio.cirillo@uniroma1.it (A.C.); francesca.defelice@uniroma1.it (F.D.F.); 8Oncology Operative Unit, Hospital of Frattamaggiore, ASL Napoli 2 Nord, Via D. Pirozzi 66, Frattamaggiore, 80027 Naples, Italy; raffaele.addeo@aslnapoli2nord.it (R.A.); pasquale.vitale3@aslnapoli2nord.it (P.V.); 9UOC Oncologia Medica, Ospedale Umberto I, Via Giuseppe Testaferrata 1, 96100 Siracusa, Italy; paolo.tralongo@asp.sr.it (P.T.); fabrizio.romano@asp.sr.it (F.R.); 10Department of Otorhinolaryngology, IRCCS San Raffaele Scientific Institute, 20132 Milan, Italy; guizzardi.mathilda@hsr.it (M.G.); giordano.leone@hsr.it (L.G.); mirabile.aurora@hsr.it (A.M.); 11Otolaryngology and MaxilloFacial Unit, INT IRCCS Foundation G Pascale, 80131 Naples, Italy; f.longo@istitutotumori.na.it

**Keywords:** HNSCC, elderly, cetuximab, radiotherapy, geriatric assessment

## Abstract

**Simple Summary:**

Elderly patients affected by head and neck squamous cell carcinomas often receive fewer intensive treatments; even though age alone does not mean frailty, this group of patients is often understudied and only few data report on the possible efficacy of cetuximab-radiotherapy combination in locally advanced head and neck carcinoma. We decided to retrospectively analyze our experience in patients over 65 years treated with this combination. We found a median overall survival comparable to what previous publications present for younger patients, with an overall compatible profile of toxicity. Our study, pending further research, suggests the possibility of using Cetuximab Radiotherapy in the elderly.

**Abstract:**

Background: A wide percentage (25–40%) of patients affected by head and neck squamous cell carcinoma (HNSCC) are over 70 years old, and they present with different characteristics if compared to younger patients. Elderly patients often receive less intensive, non-surgical, and non-multimodal treatments. Although age does not mean frailty, the elderly are at a higher risk of developing toxicity. In fact, several studies enrolling patients treated with cisplatin + radiotherapy (CISPLATIN + RT) or cetuximab + radiotherapy (Cet + RT) showed reduced efficacy over 65 years. Methods: We conducted a multicenter retrospective analysis in patients with Locally Advanced HNSCC aged over 65 years, who underwent Cet-RT, diagnosed in the period between 2017 and 2024. The primary endpoint was to describe Overall Survival (OS), the secondary endpoints were Progression Free Survival (PFS) and the percentage and type of Adverse Events (AEs). Patients received a geriatric assessment using the G8 questionnaire. Results: Data regarding Eighty-Two (82) patients were analyzed, median age was 74 years (range 65–84), most patients had oral cavity (26.8%) and laryngeal cancer (37.8%). Fifty-six point one (56.1%) of patients were smokers, and 17.1% reported alcohol consumption. All patients completed radiotherapy, and 80.5% of them developed AEs, which in 25.6% of cases were G3–4 toxicities. No relationship was found between G3–4 AEs and age (*p* = 0.596), G8score < 14 (*p* = 0.804), and smoking (*p* = 0.245)/drinking habits (*p* = 0.341). Median OS was 58 months, with a slightly non-significant positive trend in OS for patients who were non-smokers and those who did not develop G3–4 AEs (*p* = 0.786 and 0.799, respectively). Association between folliculitis and OS was statistically significant (*p* = 0.001). Conclusions: In elderly patients, Cet-RT represents a feasible, well-tolerated option, although further prospective studies are needed.

## 1. Introduction

Head and neck squamous cell carcinomas (HNSCC) represent the seventh most common cancer worldwide [1], accounting for 4% of all malignancies [2]. According to the RARECAREnet trial, 90,000 cases are diagnosed each year [3]. Patients with HNSCC are mostly male, and up to 25–40% of them are over 70 years of age [1,2]. Elderly patients present different characteristics if compared with younger patients; in fact, they more frequently present oral cavity tumors, often related to smoking, but not to alcohol consumption, and much less frequently they have HPV-related tumors [4,5,6,7,8]. Furthermore, in elderly patients, the male-to-female ratio is closer to one [5,6,7]. The majority of HNSCC diagnosed in this population is also locally advanced, although with reduced nodal extension if compared to younger groups [8,9,10].

Although standard treatment for Locally Advanced HNSCC (LA-HNSCC) is based on a combination of surgery, when feasible, and RadioTherapy (RT) with or without ChemoTherapy (CT) [11,12], Iqbal et al. reported how elderly patients frequently receive less intensive and, somewhat, suboptimal treatment when compared to younger populations. Often, treatment is non-surgical and non-multimodal [13], whereas we know how Chemo-Radioherapy (CRT) with Cisplatin or Cetuximab + RT is superior to RT alone [14,15]. On the other hand, several authors demonstrated the absence of a significant difference in survival according to age [16,17,18], especially in the elderly over 65 years of age. A possible explanation may be that other concomitant cofactors, beyond chronological age, can have an impact on prognosis.

Although we know that aging is a highly heterogeneous process and frailty does not necessarily increase with age, several reports and publications confirm that elderly patients are more frail and more susceptible to not only develop Adverse Events (AEs), but also cancer-related complications with higher frequency [17,19,20,21]. According to Daly et al. [22], discontinuation rates for elderly people tend to be higher and proportional to the age of the patient. As reported in the updated MACH-NC meta-analysis, age and Eastern Cooperative Oncology Group (ECOG) Performance Status (PS) appeared as the only significant factors influencing CRT efficacy; in fact elderly patients showed a significant shorter Overall Survival (OS) (*p* = 0.03) and a negative trend, although non-significant, in Event-Free Survival (EFS) (*p* = 0.06) [14] if compared with younger patients (<65 years old).

In 2016, Ahn et al. published a series of recommendations to discriminate patients who can be considered fit or unfit to Cisplatin-based CRT [23]. As results, low PS (>2 according to ECOG), kidney dysfunction (glomerular filtration rate < 50 mL/min), neurological deficits (grade 2 neuropathy) and reduced hearing (grade > 2 hypoacusia) are absolute contraindications to cisplatin administration, whereas age (between 65 and 75 years), comorbidities, reduced kidney (GFR between 50 and 60 mL/min) or neurological function (grade 1 neuropathy) are relative contraindications. Very elderly patients (those over 75 years old) can be considered platinum-ineligible. Indeed, the MACH-NC meta-analysis demonstrated a reduced benefit from the addition of cisplatin to RT after 50 years of age, with a lack of benefit after 71 years [7,14]. Bonner et al. demonstrated the superiority of the Cetuximab + RT combination versus RT alone in the LA setting; however, patients with oral cavity tumors were excluded, the median age of the participants was only 58 years, and the median ECOG PS was between 1 and 2. No benefit was seen in those older than 65 years [15]. Very similar results were published by Jensen et al. and Alongi et al. [24,25]. While later trials have reported that Cisplatin-Radiotherapy remains superior to Cetuximab-Radiotherapy, Cetuximab still is an alternative regimen in patients ineligible for Cisplatin. Although carboplatin as a cisplatin substitute is a frequent choice and results are superior to cetuximab, it is still burdened by a significant myelotoxicity that, in elderly patients, could influence therapy adherence and increase hospitalization risk [26,27,28,29,30,31,32]. Over the years, several predictive models of toxicity from oncological treatments have been proposed; the CARG toxicity calculator (Cancer and Aging Research Group) and the CRASH score (Chemotherapy Risk Assessment Scale for High Age Patients) were designed to identify patients who were prone to develop grade 3–5 AEs during CRT. Both systems (CRASH and CARG) indirectly underline how the advanced age of the patients can, in any case, influence the dose intensity of the oncological treatment, since alterations in multi-organ function are statistically much more frequent in elderly patients. As a consequence, several experts recommend a geriatric assessment for elderly patients addressed to oncologic treatments [27]. Different score systems have been created, but the most efficacious is the CGA (Comprehensive Geriatric Assessment), while the most practical is the Geriatric 8 (G8) [33,34]. In several trials, vulnerable patients, according to geriatric assessment, when subjected to oncological treatment, developed greater toxicity and obtained poor response to treatment with rapid deterioration of Quality of Life (QoL) and reduced OS [35,36,37,38]. With regard to patients affected by HNSCC, due to the intrinsic fragility associated with the disease caused by malnutrition, the immunosuppressed status, and the significant inflammatory response, it is crucial to understand who may be at greater risk of developing AEs from cancer treatment. A recent AIRO (Italian Association of Radiation Oncology) review reported that only 10% of multidisciplinary teams in Italy benefit from the support of a geriatrician, thus increasing the possibility of over- and under-treating patients due to wrong frailty assessment [39]. A delicate balance between intensive but not well-tolerated treatments and maintaining quality of life is difficult to obtain. Decision-making requires geriatric frailty evaluation and multidisciplinary discussion: more often than not, non-surgical monomodal treatments and/or substandard treatments are offered to elderly patients [40].

Since concomitant Cetuximab + RT is a treatment that oncologists often use in elderly patients, the present work analyzes the outcome of a group of elderly patients (>65 years old), diagnosed with LA-HNSCC, treated with this combination therapy. At the moment of our analysis (2017–2024), although results were encouraging, the use of immune checkpoint inhibitors was not allowed in LA HNSCC beyond clinical trials, thus treatment with Cetuximab was the only available choice. We collected data from different Italian centers, and we have observed the Survival and Toxicity related to the treatment.

## 2. Materials and Methods

We conducted a retrospective multicentric analysis, collecting data between 1 January 2017 and 31 May 2024, the data regarding 82 elderly (>65 years old) patients diagnosed with LA-HNSCC treated with concomitant Cetuximab + RT.

Data cut off was 20 September 2024. The data have been retrospectively collected from available clinical databases. All patients whose data were analyzed gave their consent by signing the Written Informed Consent and receiving a copy of it.

The involved centers in this analysis are: IRCSS Istituto Nazionale Tumori “Fondazione G. Pascale” (Naples), Azienda Ospedaliera Universitaria “L. Vanvitelli” (Naples) (which were also the coordinating Centers), Istituto di Ricovero e Cura a Carattere Scientifico (IRCSS) “San Raffaele” (Milan), AOU Policlinico “Umberto I”, Sapienza University of Rome (Rome), Ospedale “San Giovanni di Dio” (Frattamaggiore), and Ospedale “Umberto I” (Siracuse).

We included patients with the following features: (a) histological diagnosis of squamous cell carcinoma of the head and neck cancer originating from the following subsites: oral cavity, oropharynx, hypopharynx and larynx and squamous cell carcinoma of the head and neck with unknown primary; (b) ECOG PS scale between 0 and 2; (c) age at diagnosis over 65 years, (d) written informed consent.

All the patients in the analysis were treated with intensity-modulated radiotherapy (IMRT) with a radical intent [41,42,43], receiving a 2-Gy equivalent dose (EQD2) of 70 Gy to the high-risk regions and 54 Gy to low-risk regions. The isotropic margin from CTV to PTV was 5 mm.

Treatment with concomitant Cetuximab consisted of a loading dose of 400 mg per square meter of body surface area, followed by 6 additional maintenance doses of 250 mg per square meter of body surface area every week. All participating centers in the study administered the G8 questionnaire to patients, and for those for whom test results were available, we collected and analyzed the data. As previously mentioned, the G8 questionnaire is a useful screening tool often used in clinical practice to identify older patients with a higher risk for frailty. The G8 score assesses potential deficits in geriatric domains, including food intake, weight loss, BMI, mobility, neuropsychological condition, and polypharmacotherapy; subjective consideration by the patient is also taken into account. We used the recommended threshold of 14 to stratify our results: 14 has been identified in previous works as the cut-off that provides good sensitivity (80%) and acceptable specificity (60%) [44]. However, due to the retrospective approach to the study, the participating centers acted differently in regard to frail patients based on the report of the G8 score; thus, an analysis of the further assessments could not be carried out. All centers collected data on past medical events, cardiovascular, neurological, and pulmonary comorbidities, and current drug therapy.

The primary endpoint of our study was to describe the OS, defined as the interval between treatment start and death from any cause. Secondary endpoints were to describe the toxicities encountered by the patients during the treatment. Survival was evaluated using Kaplan–Meier estimates, whereas differences in survival were evaluated with the log-rank test, with a significance level of *p* = 0.05.

Compliance with CONSORT guidelines for retrospective studies (STROBE) was confirmed.

## 3. Results

### 3.1. Population

Ninety-one (91) patients were included in the study, 71 men and 20 women (Table 1). The median age was 74 years (range, 65–86); 40 patients were 75 years old or older at the time of treatment initiation. Twenty-five percent (23 patients) had a diagnosis of oral cavity cancer, 2.2% (2 patients) hypopharyngeal cancer, 38.5% (35 patients) laryngeal cancer, 27.5% (25 patients) oropharyngeal cancer, and 6.6% (6 patients) laterocervical metastasis from squamous cell carcinoma of unknown primary origin. Twenty-eight point six percent (28.6%) of patients were diagnosed at stage III, while 41.8% at stage IV A and 21.9% at stage IV B. Fifty-nine point three percent of patients were smokers (more than 10 pack/year), and 21.9% reported a history of alcohol consumption. Only one patient declared narcotic substance abuse. Only six patients in our cohort presented with positive p16 immunohistochemistry.

At the data cut-off, 39 patients were dead. Median number of Cetuximab cycles was 6 (range of 2–9 cycles). No patient discontinued radiotherapy due to RT-related AEs.

Seventy-five (82.5%) patients developed toxicity during or after treatment (Table 2) and were graded according to CTCAE by the primary clinician. Mainly, grade 1 or 2 side-effects were described (20.9% G1 and 33% G2, respectively). On the other hand, 26.4% of patients experienced grade 3 toxicity, and 2.2% of them suffered from grade 4 AEs; overall, 28.6% of patients developed high-grade toxicity (G3–4). The most frequently reported AEs were oral mucositis (30.8%), skin rash and radiodermitis (26.8%), folliculitis–defined as papulo-pustular acneiform rash [45] (22%), dysphagia (9.9%), xerostomia (7.7%), paronychia (5.5%), oral candidiasis (4.4%) and hypomagnesemia (1.1%) (Table 2).

No relationship was found between high degree toxicity (G3–4) and age (*p* = 0.861, OR 0.9, 95% CI 0.306–2.69), nor was any correlation observed between G3–4 AEs and low G8 score (<14) (*p =* 0.47, OR 2.27, 95% CI 0.23–22.1). No correlation between G3–4 AEs and potus (*p* = 0.067, OR 0.383, 95% CI 0.135–1.09) and between G3–4 AEs and smoking was demonstrated (*p* = 0.225, OR 1.86, 95% CI 0.678–5.09).

### 3.2. Survival and Toxicity

Median overall survival was 58 months (Figure 1).

When analyzing survival according to primary tumor site, subgroups with better survival were those with oral cavity cancer and oropharyngeal cancer. Unfortunately, no subgroup analysis for HPV status could be performed due to the limited number of HPV+ enrolled patients; thus, we could not see if HPV+ oropharyngeal cancer would fare better, as expected, than HPV-oropharyngeal cancer or oral cavity cancer. OS was then stratified according to risk factors such as smoking and alcohol. No significant benefit in survival was found between smokers and non-smokers (*p* = 0.916) (Figure 2A), although with a positive trend in non-smokers with a delta of 6 months. No statistically significant difference was found according to alcohol consumption (*p* = 0.252) (Figure 2B). A clinically meaningful difference (delta of 26 months) can be seen between patients who did not develop G3–4 AEs and those who did, although the difference was not statistically significant (*p* = 0.795) (Figure 2C). Although the difference is not statistically significant, this suggests the need for close monitoring of elderly patients and to offer supportive care and adequate management of adverse events in such a vulnerable population. A multivariate analysis was then performed (Table 3); it is worth noting that, likely due to the small sample size, the multivariate analysis suggests counterintuitively that non-smokers trend worse than smokers, and as such, these results should not be considered reliable. At the same time, the absence of statistically significant correlation between survival and smoking, alcohol, and adverse events could also be due to the small sample size.

Multivariate analysis was then conducted on different AEs. As anticipated, a statistically significant association was found between OS and folliculitis (Figure 3).

The G8 questionnaire was administered to 50 out of 91 patients. As mentioned earlier, we defined 14 as a cut-off score to identify those with worse clinical conditions [35,46]. No statistically significant relation was found between a G8 score ≤ 14 and age, G3–4 AEs, and alcohol consumption; only smoking and a lower G8 were significantly associated (*p* = 0.014), probably due to a higher risk of frailty in smokers. We performed an imputation to evaluate the associations, accounting for the missing data after confirming that these were random. After imputation, no significant difference in OS and PFS was found when stratifying for a G8 score ≤ 14, and no association was found with age, toxicity, and alcohol consumption. The statistically significant association between smoking and a lower G8 score was confirmed after imputation (*p* = 0.02). However, it must be noted that the validity of this analysis, both before and after imputation, is limited by the fact that only 55% of our cohort received a G8 questionnaire.

Twenty-nine patients (31.9%) relapsed after treatment; median progression-free survival (mPFS) was not reached, 12 months PFS was 73.6% (95% CI 63–81.6), and 24 months PFS was 67.1% (95% CI 55.9–76.0). (Figure 4), with no significant difference related to smoking, alcohol consumption, or development of grade 3–4 adverse events. The majority of patients presented with locoregional (65.5%) or lung progression (31%); remaining sites of relapse were nodes (20.7%), bone (6.9%), and liver (3.4%). Out of all patients who relapsed and for whom data about subsequent treatment are available, 27.6% were treated with systemic chemo-immunotherapy, 17.2% were treated with systemic immunotherapy alone, and 13.8% were treated with systemic chemotherapy; furthermore, 10.3% underwent a second surgery, while 20.7% continued with only palliative care and received no further treatment.

## 4. Discussion

Our retrospective analysis confirmed the feasibility of the scheme Cet-RT in elderly patients (>65 years). Although cisplatin should be the preferred choice [39,40,41] in fit patients, it is also true that the efficacy of the cisplatin-RT (Cisplatin-RT) regimen decreases with the reduction in Cisplatin dose, completely losing efficacy if the patient receives a dose lower than 200 mg per square meter of body surface [24]. In addition, in several literature reports, it seems clear that not all patients treated with concomitant three-weekly Cisplatin-RT were able to complete the three scheduled cycles: a large percentage of patients developed severe AEs, which led to a reduction in the dose-intensity of Cisplatin, compromising the effectiveness of therapy. Nakano et al., for example, really reported a 46% of systemic and locoregional toxicity in patients treated with Cisplatin-RT. [32].

Median OS in our analysis was 58 months; the data are in line with what has been seen in the literature for age-independent cohorts [32,47,48,49]. Interestingly, the mOS seen in our analysis is superior to what was observed in studies enrolling elderly patients treated with Cisplatin-RT. Haehl et al. reported a median overall survival (mOS) of 34 months [50,51], and this feature is likely due to the worse tolerability of the Cisplatin regimen in elderly patients, resulting in a lower dose of Cisplatin administered or interruption of radiation treatment. Some authors recommend Carboplatin (CBDCA) in lieu of Cisplatin for platinum-unfit patients, based on the results of some trials [40,41], which, however, were not selective in enrolling only elderly patients. In addition, another important item to be evaluated in opting for CBDCA is that it is often associated with a severe myelotoxicity risk, which could compromise therapy adherence, leading to the suspension of radiotherapy and possibly hospitalization, especially in the elderly.

In our analysis, a median of 6.5 cycles of Cetuximab were administered, and only 9 patients received fewer than 5 cycles. Four out of nine patients were over 75 years old. A third of the whole population developed G3–4, but manageable, AEs.

As expected, a negative trend in OS was seen in smokers compared with the non-smokers, and this feature is in line with the data from the literature [52,53,54]. Interestingly, smoking continuation during treatment has been associated with an increased rate of acute and late AEs, other than reduced OS and locoregional control (LRC).

In our report, OS was significantly associated with any grade of folliculitis (G1–4) (HR 0.12); a very similar finding was also found by Bonner et al., however, in that case, the association was found between OS and folliculitis of grade higher than 2 [15]. In addition, the results of a meta-analysis of 13 studies, for a total of 1961 patients with LA-HNSCC treated with concomitant Cet-RT, confirmed a significant correlation between the severity of skin rash and OS, PFS, and Overall Response Rate (ORR), independently of the site of primitive tumor [55]. According to data from the literature, acneiform rash seems to be more frequent in men, HPV-negative HNSCC, and those with lower levels of tobacco consumption [56]. Although aging is associated with immunosenescence mechanisms [57], in HNSCC, as in colorectal cancer, no reduction in cetuximab-induced dermotoxicity has been observed in the elderly [58]. In line with the above data, in our analysis, patients over 75 years old did not present significantly reduced rates of folliculitis (47% in the group < 75 years vs. 68% in the group > 75 years). On the other hand, Lugtenberg, R.T. et al. observed in a cohort of patients with HNSCC treated with cetuximab a strong correlation between drug-induced skin-toxicity and a poorer quality of life [59]. Yu, Z. et al., otherwise, demonstrated that early and rapid management of cetuximab skin toxicity, based on the use of soothing topical products, corticosteroids, and, where indicated, antibiotics, was able to improve the severity of the skin rash and its complications [60].

If compared with the results observed in the cornerstone study by Bonner et al., a slightly lower percentage of AEs has been reported in our analysis, with only a third of patients reporting mild-severe AEs, mostly cutaneous; very similar data can be found in subsequent studies [24,61]. On the other hand, Pryor, D.I. et al. reported a worse toxicity profile when compared to our analysis and to the landmark Bonner trial; however, these results were drawn from a smaller cohort, and importantly, patients were treated with non-standard or unknown radiotherapy schemes, different from IMRT, limiting comparison with other publications [62,63,64].

Although the definition of elder age is not unanimous, the threshold of 65 years old was chosen in accordance with previous works on HNSCC, where patients over the age of 65 years did not benefit from radiotherapy with cisplatin or cetuximab [14,15]. Furthermore, an age between 65 and 75 years old is defined as a relative contraindication to cisplatin according to Ahn et al. [23]. However, restricting our analysis to patients 70 years old or older did not affect the results: median OS was 58 months, with no statistically significant association between OS and smoking, alcohol consumption, or severe adverse events; a lower G8 score remains persistently associated with smoking (*p* = 0.017).

In recent years, utmost interest has been pointed in designing clinical trials based on the combination of Immune Checkpoint Inhibitors (ICI) and cetuximab in LA-HNSCC. Overall, the results were initially disappointing, but finally, the tables have turned. In the GORTEC 2017-01 REACH trial, the addition of Avelumab to Cet-RT was associated with higher PFS and distant control rates, although no benefit in OS was reported [65,66]. Significantly, both Pembrolizumab and Nivolumab have been studied in a neoadjuvant or curative setting, increasing disease-free survival and event-free survival; however, the use of immune checkpoint inhibitors beyond the setting of the REACH trial remains limited to combination with Cisplatin, thus restricting their use in elderly, frail patients [66,67,68,69].

The main limitations of our report are, first, its retrospective nature, which increases the risk of information and recall bias, and second, the high incidence of incomplete data. Patients were not actively selected for their adherence to treatment; however, no patient in our cohort discontinued radiotherapy, which may impact the generalizability of our results, as this is relatively uncommon. G8 data were available for only 55% of our patients; this finding was independent of the participating center or patient age. This severely limits our findings and needs prospective and more comprehensive trials on the role of frailty to possibly corroborate our findings. Moreover, the relatively small number of patients studied (less than 100) limited the possibility of carrying out subgroup analyses, considering, for example, the subgroup of HPV-related patients, different sub-sites, or different TNM stages.

## 5. Conclusions

In elderly platinum-unfit patients, Cetuximab + RT treatment represents a valid option, associated with a reasonable survival rate at the cost of mostly manageable toxicity. Skin toxicity should be addressed early and then reduced in severity and duration. This could positively impact patient adherence to treatment. Using CBDCA in platinum-unfit patients does not reduce the risk of myelotoxicity, which, particularly in elderly patients, could promote the onset of serious infections, aspiration pneumonia, and severe mucositis. Immunotherapy, while promising, remains at the moment limited to clinical trials, at least in Italy. While we are in need of prospective trials, with a larger cohort and geriatric assessment data for all patients, and biomarker research to better select patients, we strongly recommend, also in light of the results observed in this retrospective analysis, using cetuximab as the first-choice therapy in conjunction with radiotherapy in platinum-unfit patients and in elderly patients (Table 4).

## Figures and Tables

**Figure 1 cancers-17-03550-f001:**
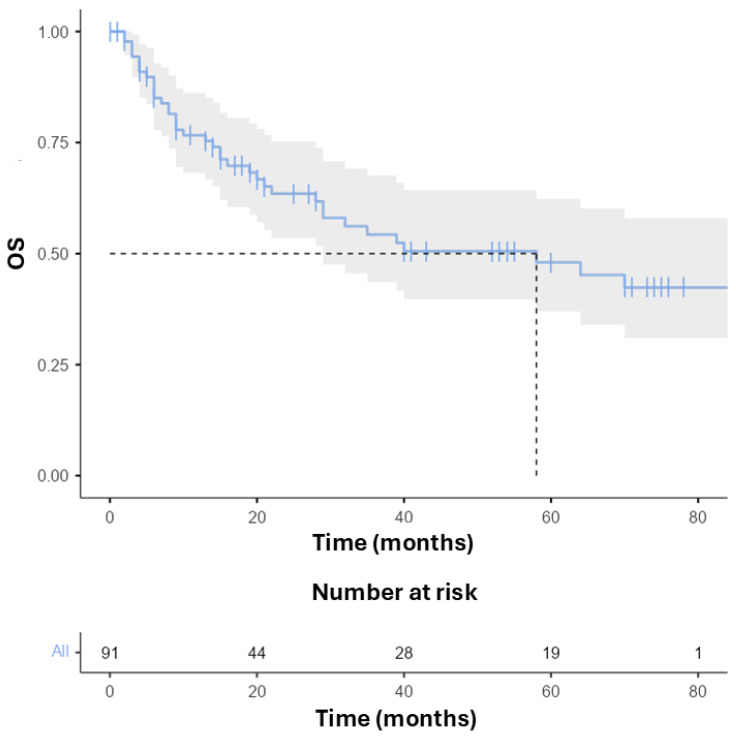
Overall survival (OS) Kaplan–Meier in the whole population.

**Figure 2 cancers-17-03550-f002:**
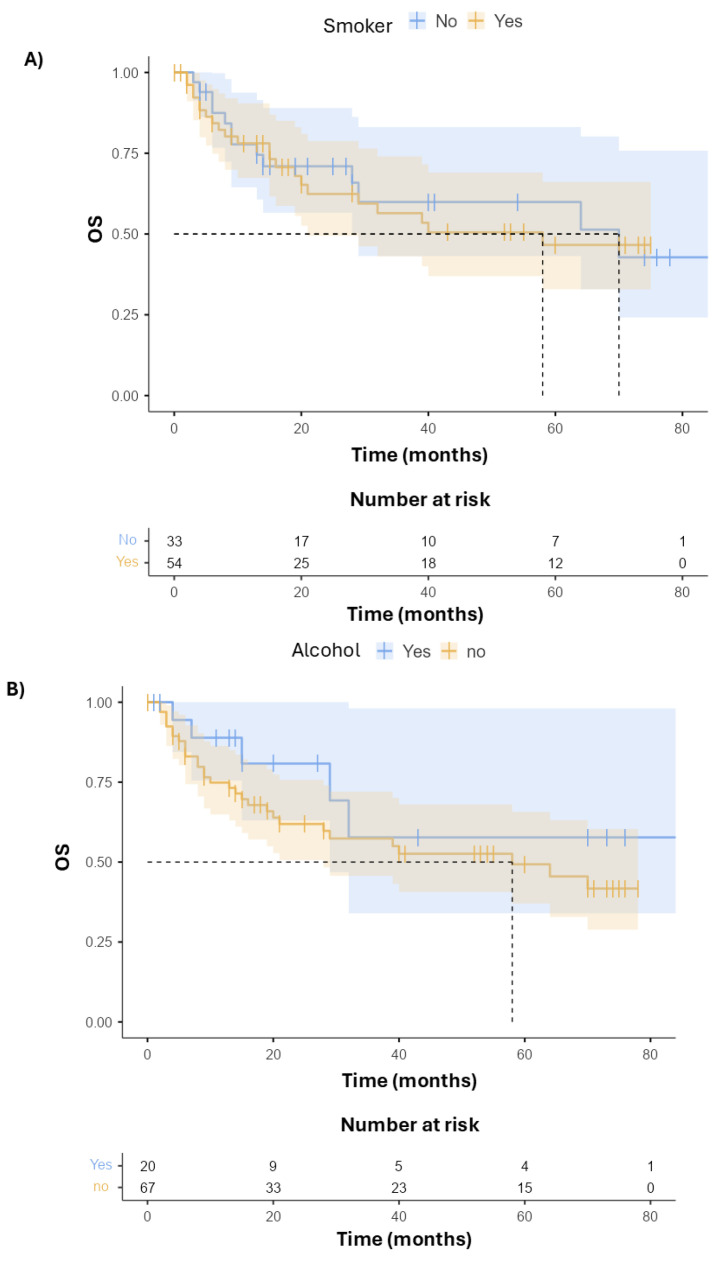
(**A**) Overall survival (OS) Kaplan–Meier stratified per smoking, (**B**) alcohol consumption (**C**) G3–4 adverse events.

**Figure 3 cancers-17-03550-f003:**
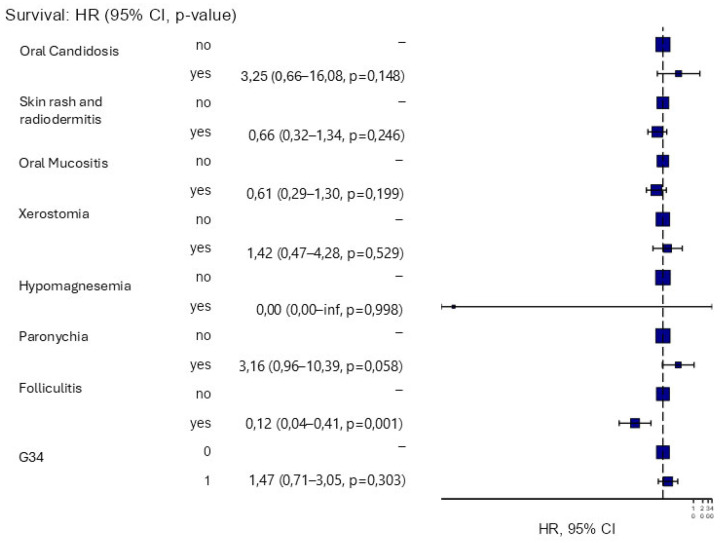
Forest plot of the association between toxicity and survival according to grade and type of AEs.

**Figure 4 cancers-17-03550-f004:**
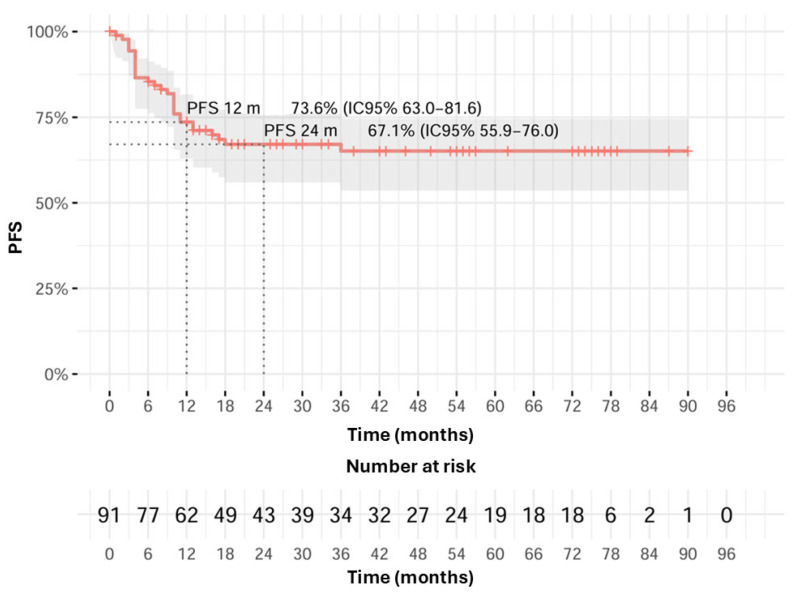
Progression-free survival Kaplan–Meier curve.

**Table 1 cancers-17-03550-t001:** Population characteristics.

Population	82	
*Females*	18	22%
*Males*	64	78%
**Median age**	74	(65–89 years)
**Median G8 score**	11.6	(4–17)
**Primary tumor sites**		
*Oral cavity*	23	25.38%
*Hypopharynx*	2	2.2%
*Larynx*	35	38.5%
*Oropharynx*	25	27.5%
*Unknown primary*	6	6.6%
**Stage**		
*III*	26	28.6%
*IVa*	38	41.8%
*IVb*	20	21.9%
*Unknown*	7	7.9%
**Smokers**	54	59.3%
**Alcohol consumption**	20	21.9%
**Substance abuse**	1	1.1%

**Table 2 cancers-17-03550-t002:** Adverse events.

Adverse Event	All Grade	G3–4
Skin rash and radiodermatitis	26.8%	9.9%
Oral Mucositis	30.8%	15.5%
Folliculitis	22%	5.5%
Dysphagia	9.9%	6.6%
Xerostomia	7.7%	2.2%
Paronychia	5.5%	0%
Oral Candidiasis	4.4%	0%
Hypomagnesemia	1.1%	0%

**Table 3 cancers-17-03550-t003:** Multivariate analysis of overall survival.

			HR (Univariable)	HR (Multivariable)
Smoking	Yes	10 (26.3)	-	-
	No	28 (73.7)	1.39 (0.29–6.56, *p* = 0.678)	3.14 (0.54–18.14, *p* = 0.201)
Alcohol	No	11 (28.9)	-	-
	Yes	27 (71.1)	0.00 (0.00–Inf, *p* = 0.999)	0.00 (0.00–Inf, *p* = 0.998)
G3–4 AEs	No	6 (15.8)	-	-
	Yes	32 (84.2)	0.81 (0.17–3.82, *p* = 0.788)	0.67 (0.12–3.90, *p* = 0.656)

**Table 4 cancers-17-03550-t004:** Contraindications and factors to consider in choosing the Cetuximab + RT regimen.

**Absolute Contraindications**	**Notes**	**References**
Impaired renal function	No data on patients with Creatinine ≥ x1.5 upper limit value	[70]
Severe marrow dysfunction	No data on patients with Hb < 9, Leukocyte < 3000, Neutrophils < 1500, platelets < 100,000	[70]
Hepatic dysfunction	No data on patients if AST/ALT ≥ x5 upper limit value Case report of Cetuximab-related DILI	[70,71]
Respiratory dysfunction	Increased risk of ILD	[72,73]
Metabolic dysfunction	Risk of hypomagnesemia, hypocalcemia	[70,74]
**Relative Contraindications**	**Rationale**	**References**
Age	No clear benefit over 65 years old of Cetuximab + RT in LA HNSCC	[15,24,25]
Performance status	Patients presented PS ≤ 2 in the Bonner trial of Cetuximab + RT	[15]
Weight loss	Risk of increased weight loss due to mucositis, diarrhea	[25,75]
Cardiovascular dysfunction	Increased frequency of severe cardiovascular events	[70]
Intercurrent infections	Risk of secondary infections of cutaneous and mucosal lesions	[70,76]
**Intrinsic Factors to Consider**	**Rationale**	**References**
Smoking	Risk of increased resistance, ILD, and toxicity	[73,77,78,79]
Pre-existing dermatological conditions	Worsening of pre-existing dermatological lesions	[76,80,81]
Sun exposure	Sun sensitivity, no benefit from sunscreen	[76,82]
Hearing function	Increased hearing loss with CisRT than with Cetuximab + RT	[83]

## Data Availability

The data presented in this study are available on request from the corresponding author due to privacy concerns.

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
