# Peer review of "Retrospective Trial on Cetuximab Plus Radiotherapy in Elderly Patients with Head and Neck Squamous Cell Cancer"

_cancers, 2025, doi:10.3390/cancers17213550_

Round 1

Reviewer 1 Report (New Reviewer)

Comments and Suggestions for Authors

This is a retrospective study of the clinical outcomes of elderly patients with HNC receiving cetuximab concurrent with RT. Patient QOL was rightly included prominently in the Results.

Comments

  1. While the definition of "elderly" may differ, it usually refers to people older than 70. You included patients older than 65. Please add a brief  analysis of outcomes and toxicities confined to those whose age was  >70 year
  2. 2. Folliculitis is not a common side effect in the therapy of HNC. Please provide a description of what exactly defines it
  3. You may add to Introduction the fact that concurrent chemotherapy was not found to be beneficial in patients aith HNC older than 70 years in the metaanalyses 
  4. 4. Please review potential typos. For exaple, "Noe" rather than "No" on page 5
Comments on the Quality of English Language

needs review of potential typos

Author Response

  1. While the definition of "elderly" may differ, it usually refers to people older than 70. You included patients older than 65. Please add a brief  analysis of outcomes and toxicities confined to those whose age was  >70 year

As reported by the reviewer, the definition of elderly differs based on the chosen classification. However, we choose this cut-off for a number of reasons: EMA and NIH use 65 years old as the cut off; 65 years is also used as cut off in the MACH-NC meta-analysis (where older patients present with shorter OS) and in the Cetuximab+RT trial by Bonner (where pts > 65 years old reported no benefit from CetRT); age between 65 and 75 years old also represent a relative contraindications to cisplatin as defined by the Ahn criteria.

Nevertheless, results were confirmed even when restricting analysis to those 70 years old or older, with a mOS of 58 months, no statistically significant association of OS and G3-4 AEs, smoking or alcohol, and a statistically significant association between a lower G8 score and smoking (p 0.017).

  1. Folliculitis is not a common side effect in the therapy of HNC. Please provide a description of what exactly defines it
    A definition has been added, also referring a previous published work by Bonomo et al (10.1016/j.critrevonc.2017.10.011) that analyses the role of folliculitis and acneiform rash in HNSCC treated with Cetuximab.
  2. You may add to Introduction the fact that concurrent chemotherapy was not found to be beneficial in patients aith HNC older than 70 years in the metaanalyses
    This mention has been added in the second paragraph of the Introduction.
  3. Please review potential typos. For exaple, "Noe" rather than "No" on page 5
    Corrected this and other typos.

Reviewer 2 Report (New Reviewer)

Comments and Suggestions for Authors

The study addresses a clinically relevant and challenging population elderly patients with locally advanced Head and Neck Squamous Cell Carcinoma who are often undertreated due to comorbidities and frailty. The use of cetuximab and radiotherapy as a treatment alternative to cisplatin-based chemoradiation is a vital area of research. While the data collection and clinical analysis are solid, several key methodological details are missing, the statistical interpretation is occasionally weak, and the presentation of the frailty assessment is deficient.

  • The methods section provides a good overview but lacks critical detail necessary for reproducibility and interpretation. For instance, the abstract and results frequently discuss frailty and the G8 score as primary risk factors. However, the Methods section is silent on how comorbidities (beyond the G8 score) were assessed and quantified (e.g., charlson comorbidity index, ASA score). The G8 score is an excellent frailty screening tool, but it should be reported as a continuous variable alongside its median and range, as done in Table 1, and its components should be described in the Methods.
  • The authors should explicitly state in the Methods what the G8 questionnaire measures and why the cut-off of 14 was chosen for statistical analysis (Line 39).
  • The text states, "only 55% of our cohort received a G8 questionnaire" (Line 45). This is a substantial limitation (∼45% missing data) for the key risk factor of frailty. The authors must discuss the implications of this missing data. Was the missing data random? Did they perform a sensitivity analysis or any imputation methods (even if simple) to test the robustness of the G8-related findings?
  • The treatment is described as intensity-modulated radiotherapy with a dose of in For a definitive retrospective trial, the authors must specify the CTV (Clinical Target Volume) and PTV (Planning Target Volume) margins and the dose prescription to the primary tumor and involved nodes (e.g., 70 Gy to high-risk PTV, 56 Gy to low-risk PTV).
  • The Results (L58-59) state, "subgroups with better survival were those with oral cavity cancer and oropharyngeal cancer." This finding is unexpected, as oral cavity generally has a poorer prognosis than -positive oropharyngeal cancer. This claim should be moderated and discussed in the context of status, which is not mentioned in the patient characteristics (Table 1).
  • The univariable for smoking is with =0.678 (Table 3), meaning a non-smoker trended toward worse survival, which is counter-intuitive and may be a statistical fluke of the small sample size.
  • The authors must explicitly state that the lack of statistical significance for smoking, alcohol, and is likely due to the small sample size (=82).
  • The text highlights a "statistically significant association was found between OS and folliculitis" (Line 35) and OS and Oral Candidosis (Line 33).
  • Looking at the Forrest Plot (Figure 3), the for Folliculitis is (), and the -value is . This is not conventionally statistically significant (<0.05). Oral Candidosis is (), =0.148, also not significant.
  • The claim of a "statistically significant association" must be corrected to "a strong positive trend" or "borderline significance," especially for folliculitis (=058). Avoid overstating the result; the for Folliculitis is high, but the wide suggests low power.
  • The incidence of G3-4 Adverse Events (AEs) is relatively high at 25.6% (Abstract, L20), confirming the toxicity concern in the elderly. However, the text (Line 41) suggests an association between G3-4 AEs and a 26-month difference in OS on the Kaplan-Meier curve (Figure 2C), but the p-value is 0.795.
  • The authors should focus less on the non-significant -value and more on the clinical significance of the -month difference, arguing that this large delta, even if non-significant, reinforces the need for close monitoring in this vulnerable population.

Author Response

  • The methods section provides a good overview but lacks critical detail necessary for reproducibility and interpretation. For instance, the abstract and results frequently discuss frailty and the G8 score as primary risk factors. However, the Methods section is silent on how comorbidities (beyond the G8 score) were assessed and quantified (e.g., charlson comorbidity index, ASA score). The G8 score is an excellent frailty screening tool, but it should be reported as a continuous variable alongside its median and range, as done in Table 1, and its components should be described in the Methods.
    The requested information on the G8 questionnaire was added in the methods section. However, due to the retrospective studies, different centres acted differently in regard to subsequent assessments and thus no other analysis has been performed. This has been specified in the manuscript.
  • The authors should explicitly state in the Methods what the G8 questionnaire measures and why the cut-off of 14 was chosen for statistical analysis (Line 39). Text has been edited accordingly; the reason behind cut-off decision, present in the Results, has been explained also in the methods section.
  • The text states, "only 55% of our cohort received a G8 questionnaire" (Line 45). This is a substantial limitation (∼45% missing data) for the key risk factor of frailty. The authors must discuss the implications of this missing data. Was the missing data random? Did they perform a sensitivity analysis or any imputation methods (even if simple) to test the robustness of the G8-related findings? The missing data were random, we added this bias and this explanation in the bias paragraph in the Results. No further analysis was performed.
  • The treatment is described as intensity-modulated radiotherapy with a dose of 70 Gy in 35 For a definitive retrospective trial, the authors must specify the CTV (Clinical Target Volume) and PTV (Planning Target Volume) margins and the dose prescription to the primary tumor and involved nodes (e.g., 70 Gy to high-risk PTV, 56 Gy to low-risk PTV). Information has been added.
  • The Results (L58-59) state, "subgroups with better survival were those with oral cavity cancer and oropharyngeal cancer." This finding is unexpected, as oral cavity HNSCC generally has a poorer prognosis than HPV-positive oropharyngeal cancer. This claim should be moderated and discussed in the context of HPV status, which is not mentioned in the patient characteristics (Table 1). Text has been edited accordingly.
  • The univariable HR for smoking is 39 with p=0.678 (Table 3), meaning a non-smoker trended toward worse survival, which is counter-intuitive and may be a statistical fluke of the small sample size. Text has been edited accordingly.
  • The authors must explicitly state that the lack of statistical significance for smoking, alcohol, and G3-4 AEs is likely due to the small sample size (N=82). Text has been edited accordingly.
  • The text highlights a "statistically significant association was found between OS and folliculitis" (Line 35) and OS and Oral Candidosis (Line 33). Looking at the Forrest Plot (Figure 3), the HR for Folliculitis is 16 (95% CI: 0.96-10.39), and the p-value is 0.058. This is not conventionally statistically significant (p<0.05). Oral Candidosis is 3.25 (95% CI: 0.66-16.08), p=0.148, also not significant. The claim of a "statistically significant association" must be corrected to "a strong positive trend" or "borderline significance," especially for folliculitis (p=058). Avoid overstating the result; the HR for Folliculitis is high, but the wide CI suggests low power.
    The folliculitis p is p = 0.001 (p = 0.058 is related to paronychia, which is not suggested to be statistically significant).
  • The incidence of G3-4 Adverse Events (AEs) is relatively high at 25.6% (Abstract, L20), confirming the toxicity concern in the elderly. However, the text (Line 41) suggests an association between G3-4 AEs and a 26-month difference in OS on the Kaplan-Meier curve (Figure 2C), but the p-value is 0.795. The authors should focus less on the non-significant p-value and more on the clinical significance of the 26-month difference, arguing that this large delta, even if non-significant, reinforces the need for close monitoring in this vulnerable population. Text has been edited accordingly.

Round 2

Reviewer 2 Report (New Reviewer)

Comments and Suggestions for Authors

The authors have substantially improved the manuscript in response to the previous round of my comments. Most of the critical issues have been addressed and two points remains:

  • While the authors state that retrospective variability across centers prevented consistent use of indices such as Charlson or ASA, the absence of any quantification of comorbidities is a notable limitation. At minimum, a brief description of what (if any) comorbidity information was collected at participating centers would help.
  • The authors explain missingness as random and acknowledge bias in Results. However, no sensitivity analysis or imputation was attempted. Even if imputation is not feasible, the authors could strengthen the discussion by explicitly stating why and outlining how this limits the robustness of the findings.

Author Response

  • While the authors state that retrospective variability across centers prevented consistent use of indices such as Charlson or ASA, the absence of any quantification of comorbidities is a notable limitation. At minimum, a brief description of what (if any) comorbidity information was collected at participating centers would help. Text was edited accordingly, explaining which comorbidities were collected in all centres.
  • The authors explain missingness as random and acknowledge bias in Results. However, no sensitivity analysis or imputation was attempted. Even if imputation is not feasible, the authors could strengthen the discussion by explicitly stating why and outlining how this limits the robustness of the findings. Imputation was done and added to the results, although we stated also that the percentage of missing data limits our results.

This manuscript is a resubmission of an earlier submission. The following is a list of the peer review reports and author responses from that submission.

Round 1

Reviewer 1 Report

Comments and Suggestions for Authors

The manuscript presents a retrospective analysis of Cetuximab plus Radiotherapy (CetRT) in elderly patients with locally advanced HNSCC.

While the topic is clinically relevant, the study has several weakness which means the manuscript is not ready for publication at a prestigious journal such as Cancers MDPI.

However, I provide detailed and constructive feedback below tp help the authors improve their work.

  1. The abstract lacks doesn’t include statistical details (e.g., confidence intervals for OS, p-values for survival comparisons).
  2. The statement "no relationship was found between G3-4 AEs and age, G8<14, smoking/drinking" is vague without statistical metrics.
  1. Clarify whether the "positive trend" for non-smokers was statistically significant.
  1. The conclusion is very optimistic given the retrospective design and small sample size. Tone down the conclusion to reflect the limitations (e.g., "CetRT may be a feasible option, but further prospective studies are needed").
  2. The introduction is weak and needs to be strengthened. The logic for focusing on CetRT in elderly patients is not contrasted with current alternatives enough (e.g., immunotherapy, carboplatin-based regimens).
  3. The sentence that "age alone is a limiting factor to platinum fitness" is not fully supported by recent literature (e.g., Sun et al. 2022, Blanchard et al. 2023).
  4. There’s no mention of the limitations of Cetuximab (e.g., inferior survival compared to platinum in some studies).
  5. Mention why immunotherapy was not considered in this cohort, given its growing use in LA-HNSCC.
  6. There are some weaknesses in the Methods as well. Includes mixed primary sites (oral cavity, larynx, etc.) and stages (III-IVB), which may confound outcomes.
  7. G8 was only administered to 42/82 patients (51%), which can affect its validity as a frailty marker.
  8. Dose constraints, organ-at-risk metrics, and IMRT quality assurance are not described.
  1. Include radiotherapy protocol details (e.g., dose-volume histograms, OAR sparing).
  2. Report compliance with CONSORT guidelines for retrospective studies (STROBE).
  3. The Kaplan-Meier curves lack clarity (e.g., no numbers at risk, unclear censoring). The "delta of 25 months" for G3-4 AEs is descriptive without statistical significance (p=0.799).
  4. CTCAE grading is mentioned, but no details on how AEs were adjudicated (e.g., clinician-reported vs. patient-reported).
  5. The significant association with smoking (p=0.004) is not contextualized (e.g., is smoking a proxy for frailty?).
  6. The conclusion that CetRT is "well-tolerated" is not really supported (25.6% G3-4 AEs is substantial).
  7. The discussion doesn’t have critical engagement with studies showing inferiority of CetRT to platinum (e.g., Beckham et al. 2020).
  8. The biological plausibility of this association is not discussed (e.g., is it a marker of EGFR inhibition or immune response?).
  9. Mention the limitations in a separate paragraph including retrospective design, small sample, missing data, lack of PFS analysis.
  10. Only 6 patients were p16+, yet HPV status is a critical prognostic factor in HNSCC. This should be discussed as a limitation.
  11. The conclusion has serious weaknesses and it overstates CetRT's efficacy ("valid and well-tolerated") without addressing its inferiority to platinum in recent studies.
  12. Reframe to emphasize CetRT as an option for platinum-unfit patients, pending better alternatives (e.g., immunotherapy combinations).
  13. The conclusion doesn’t mention the need for prospective validation or biomarker-driven patient selection.
  14. Call for prospective trials with geriatric assessments and biomarker stratification.
  15. Figures 1, 2, 3 and 4 should be combined into 1 figure (A, B, C, D)

Reviewer 2 Report

Comments and Suggestions for Authors

Dear authors,

I reviewed your study, and stopped after reading the methods and results section. There are substantial deficits in the design, analysis and reporting of this study and I recommended rejecting it. I am attaching my comments and hope you get this study published somewhere else after substantial improvements and revision.

Kind regards,

  1. Abstract: please explain the abbreviations "HNSCC" and "CDDP+RT and Cet+RT" before using them.
  2. Several lines in the abstract are not full sentences !! For example, Line 35:  "primary endpoint OS", and line 36: 82 patients .. to alcohol consumption.
  3. What's mOS in Line 39? Please explain all abbreviations before using them.
  4. Please include the abbreviations in the text. There is no need to list them in the end of the study, and it's not the journal's style. Is it necessary to abbreviate cisplatin of cetuximab? It's only one word !

Methods:

  1. I find it very unplausible that not even one patient discontinued his RT treatment ! There is definitely a patients' selection, which was not reported adequately in the manuscript. Selecting only patients who finished their treatment, would shows an unrealistic survival. Was treatment completion an inclusion criteria?
  2. Line 122: the term "from classical district" is unheard of. Please use "from the following subsites".
  3. Why wasn't duration to recurrence considered as a relevant outcome?
  4. I don't see a reason, not to conduct a cox regression with the available survival data. 

Results:

  1. It doesn't make sense to examine the association between AEs and OS. The observed effect with folliculitis is a mere chance. First the question of this study is not the association of AE with survival in elderly patients undergoing irradiation, but the effect of age and frailty in the elderly receiving RT and cetuximab. The authors should examine the effect of Age, ECOG status, comorbidities, irradiation dose, BED and immuntherapy on survival and disease recurrence. Disease recurrence was not mentioned in the study design. 
    AEs - while important - are not the main exposure of the study. They were suddenly presented in the results without proper presentation of the statistical models used to examine their correlation with OS in the methods section. Then coincidence results are interpreted as causally! This is not correct.

Tables:

  1. in Table 1 please replace POTUS with alcohol consumption.
  2. I urge the authors to improve the format of Table 1. It is possible to write full words instead of "F" and "M".

Plots:

  1. I don't see why plotting survival by smoking, alcohol consumption or AE is interesting. The exposure of the study is age; Other variables that reflect frailty such ECOG score are also important. I urge the authors to consider those when plotting the Kaplan-Meier curves.